# Influential Path of Social Risk Factors toward Suicidal Behavior—Evidence from Chinese Sina Weibo Users 2013–2018

**DOI:** 10.3390/ijerph18052604

**Published:** 2021-03-05

**Authors:** Yujin Han, He Li, Yunyu Xiao, Ang Li, Tingshao Zhu

**Affiliations:** 1Institute of Psychology, Chinese Academy of Sciences, Beijing 100101, China; hanyujinruc@163.com (Y.H.); lih@psych.ac.cn (H.L.); 2Department of Statistics, Renmin University of China, Beijing 100872, China; 3Department of Psychology, University of Chinese Academy of Sciences, Beijing 100049, China; 4Silver School of Social Work, Indiana University–Purdue University Indianapolis, Indianapolis, IN 46202, USA; yx18@iu.edu; 5Department of Psychology, Beijing Forestry University, Beijing 100083, China; angli@bjfu.edu.cn

**Keywords:** suicidal behavior, social media, path analysis model

## Abstract

(1) Purpose: The purpose of this study was to determine suicidal risk factors, the relationship and the underlying mechanism between social variables and suicidal behavior. We hope to provide empirical support for the future suicide prevention of social media users at the social level. (2) Methods: The path analysis model with psychache as the mediate variable was constructed to analyze the relationship between suicidal behavior and selected social macro variables. The data for our research was taken from the Chinese Suicide Dictionary, Moral Foundation Dictionary, Cultural Value Dictionary and National Bureau of Statistics. (3) Results: The path analysis model was an adequate representation of the data. With the mediator psychache, higher authority vice, individualism, and disposable income of residents significantly predicted less suicidal behavior. Purity vice, collectivism, and proportion of the primary industry had positive significant effect on suicidal behavior via the mediator psychache. The coefficients of harm vice, fairness vice, ingroup vice, public transport and car for every 10,000 people, urban population density, gross domestic product (GDP), urban registered unemployment rate, and crude divorce rate were not significant. Furthermore, we applied the model to three major economic development belts in China. The model’s result meant different economic zones had no influence on the model designed in our study. (4) Conclusions: Our evidence informs population-based suicide prevention policymakers that incorporating some social factors like authority vice, individualism, etc. can help prevent suicidal ideation in China.

## 1. Introduction

Suicide is a serious global public health problem. Nearly 800,000 people commit suicide every year, and more people have suicidal thoughts [1]. In China, about 100,000 people die from suicide every year [2]. Therefore, it is necessary to identify important risk factors and mechanisms that link suicidal behaviors in order to design effective suicide prevention programs.

Suicidal behavior is considered to result from a complex interaction among biological, psychological, and social variables [3,4]. Many researchers have designed studies to discuss the influence of social factors on suicidal behavior, but there have been deficiencies. (1) The first is a lack of data. Relevant research has mostly relied on case studies, regional comparisons, field investigations, individual cases, and other qualitative methods to summarize the impact of social variables on suicidal behavior [5,6]. However, the sizes of the samples from these studies have been relatively small, so the conclusions may not be generalizable. (2) They have mainly been cross-sectional studies [7,8,9]. This means it has been difficult to make causal inferences. (3) Few psychological factors have been considered as mediating mechanisms. Some scholars have analyzed the relationship between suicide rates and social variables based on a large amount of data using time series models [10], but such studies have often focused on quantitative relationships while ignoring psychological factors and the influence of social factors on suicide. The stress-susceptibility model has shown that it is necessary to consider the psychological factors of suicide [11,12].

To solve the shortcoming of the lack of data, this study used large social media data and official statistics to examine the quantitative relationships between social variables and suicidal behavior. The data were obtained from the National Bureau of Statistics and Sina Weibo (a Chinese version of Twitter) in 2013–2018. Compared with the traditional methods, Sina Weibo can provide us with a large amount of data and a new research perspective. Sina Weibo is the largest microblog platform in China. Users can post text, pictures, or videos to share their lives and opinions in real time. Related studies have pointed out that some people tend to choose social media to express their abnormal emotions [13]. Thus, the social media platform is convenient for the study of suicidal behavior. Recently, social media such as Sina Weibo have acted as a common platform to conduct suicide research. In 2017, Liu et al. [14] proposed to actively identify potential suicides based on social media. Tian et al. [15] used deep learning models to predict suicide on Weibo in 2018. These studies prove the feasibility and rationality of suicide research through Sina Weibo.

In the meantime, we used the path analysis model and introduced the mediator, psychache, into our model. We hypothesized that the underlying mechanism of suicidal behavior and social factors was that social factors induced negative emotions (e.g., psychache), and negative emotions led to suicidal behavior. A negative emotion like psychache was the mediating factor; social factors cannot directly lead to suicidal behavior. We made the hypothesis based on the “stress susceptibility” model [11]. Suicide is supposed to be the result of the interaction between stress variables and individual quality. “Stress” refers to a state of tension caused by changes in dangerous or unexpected external conditions. When an individual encounters stressing events, this may lead to disorders and negative emotions. In addition, social variables can only be regarded as negative life events that are a kind of “stress source” for suicidal behavior [16,17], rather than negative emotions. “Stress sources” cannot directly led to suicidal behavior. Thus, we speculated that social factors need to lead the emergence of disorders and negative emotions, and then make individuals commit suicide. We chose psychache as the representative of negative emotions. This is because according to Shneidman’s theory [12], as a negative emotion, psychache (negative hurt, mental pain, soreness) is the cause of all suicidal behaviors, and other psychological variables (such as depression) are only related to psychache [12]. Shneidman’s theory has been confirmed by a large number of studies [18,19]. We can show the above analysis in Figure 1.

## 2. Method

### 2.1. Participants and Data Collection

First, we collected Weibo posts to calculate their suicidal behavior and mediator, psychache, as measured by suicidal behavior and psychache in the Chinese Suicide Dictionary. We downloaded and extracted all public micro blogs of Chinese mainland users during 2013 to 2018 through the application programming interface (API). Then, from the data pool, we selected Sina Weibo users living up to the following criteria: (1) Registered before January 2013 (2) Authentication type was non-institutional and non-business. (3) Had posted at least one original Weibo per month from 2013 to 2018. The final data included 4 billion micro blogs, involving 1,160,000 Sina Weibo users and 31 provincial administrative regions of Chinese mainland (excluding Hong Kong, Macao, and Taiwan). After that, we employed the “TextMind” system developed by the Computational CyberPsychology Laboratory at the Institute of Psychology, Chinese Academy of Sciences, to process the word segmentation of Weibo posts and calculate word frequency of target words in each provincial administrative region every year. Ethical approval of the research was obtained and the ethics code is H15009, approved by the Research Ethics Committee of the Institute of Psychology, Chinese Academy of Sciences. In addition, we described the Chinese Suicide Dictionary in Section 2.1.1. 

Then, we artificially divided social factors that have been confirmed to be related to suicidal behavior into three categories. These indicators focus on different aspects of social factors. They include cultural/value factors, economic factors, and other factors. 

For cultural/value factors, we collected variables identified as suicide risks in the previous literature, including individualism and collectivism [20], social morality [21]. We calculated the word frequencies of cultural/value factors based on Cultural Value Dictionary and Moral Foundation Dictionary. We described the Cultural Value Dictionary and Moral Foundation Dictionary in Section 2.1.2 and Section 2.1.3. We found the target words from dictionaries mentioned above, and calculated the defective words reflecting collectivism, individualism, and social morality in a similar way to calculating the frequency of suicidal behavior and psychache.

For economic factors, we collected variables identified as suicide risks in the previous literature. They were gross domestic product (GDP) and disposable income of residents [22]. Data of disposable income of residents and GDP from National Bureau of Statistics were used in the study.

For other factors, we collected variables identified as suicide risks in the previous literature, including urbanization [23,24,25], unemployment [26], marriage [27]. More specifically, three indicators were selected to quantify urbanization. First, according to the definition of urbanization, the analysis chose the ratio of the growth value of the primary industry to the regional GDP as an indicator to measure the proportion of the primary industry in urbanization. Second, according to the research of Wang et al., the urban population density, and public transport and car for every 10,000 people were taken as the index of population urbanization and social urbanization [28]. All data were from National Bureau of Statistics.

#### 2.1.1. Chinese Suicide Dictionary

The words reflecting suicidal behavior and psychache were selected from Chinese Suicide Dictionary. Meizhen LV et al. [29] published the Chinese Suicide Dictionary in 2015, which contained 13 dimensions of suicide expression, a total of 2168 words. The details of the dictionaries are shown in Table 1.

#### 2.1.2. Cultural Value Dictionary 

The Cultural Value Dictionary is regarded as the quantification of individualism and collectivism. Based on the cross-cultural research of individualism and collectivism, and the help of expert group discussion, the Chinese words related to individualism and collectivism are finally determined [30]. There are 53 words of individualism, such as “I”, “competition”, and 64 words of collectivism, such as “we” and “dedication”. The details of the dictionaries are shown in Table 2.

#### 2.1.3. Moral Foundation Dictionary

The Moral Foundation Dictionary is a measure of social morality. Based on the basic theory of morality, the Chinese Revision of the dictionary of morality was carried out [31,32]. According to the two aspects of morality, that is, “Virtue” and “Vice”, the amendment divides the five foundations of moral theory (care, fairness, loyalty, authority, holiness) into 10 dimensions, positive and negative. The specific “Virtue” includes purity virtue, authority virtue, ingroup virtue, fairness virtue and harm virtue. The corresponding “Vice” includes five dimensions: purity vice, authority vice, ingroup vice, fairness vice and harm vice. In order to avoid too many variables increased the complexity of our model, only negative dimensions of the dictionary (purity vice, authority vice, ingroup vice, fairness vice and harm vice) were used to analyze the impact of social morality on suicidal behavior. The details of the dictionaries are shown in Table 3.

### 2.2. Data Analysis

In our data analysis, the null value in the data was removed. We took the “year” and the “province (or autonomous region)” as the granularity and finally formed 6 (Time) and 31 (Cross-section), a total of 186 sets of data. Moreover, in order to adjust the dimensions of different variables, the original data was standardized on the basis of standard deviation by Z-score. Then, according to the “stress susceptibility” model [11] and Shneidman’s theory [12], we designed the path analysis model with mediation psychache. This means that in our model, 14 social variables only affected suicidal behavior through psychache. In addition, for assessment of path analysis model fit, we used the root mean square error of approximation (RMSEA), *the*
*chi*-*square*/*DOF*
*ratio* (χ^2^/df), Bentler-Bonett normed fit index (NFI) and comparative fit index (CFI). Acceptably, a CFI and NFI greater than 0.90, a RMSEA less than 0.10 and *a chi*-*square*/*DOF*
*ratio* between 1 and 3 indicate adequate model fit. Finally, we presented the results of the path analysis model by showing standardized coefficients and goodness-of-fit statistics. All maximum-likelihood estimations for the model were computed using R (Version 1.3.959). 

## 3. Results

We described the statistical characteristics of our original data and presented bivariate correlations after standardization among the variables in our model (Table 4 and Figure 2). There were 7 individual variables and 5 social variables with large dimensional differences, which meant we need to make standardized processing of the original data collected. The correlation coefficient thermodynamic diagram showed that the independent variables harm vice and collectivism had strong correlation (ρ>0.8). As for whether the two variables overlapped, we could further consider whether to modify the model according to the goodness of fit statistics of the path analysis model [33]. The descriptive statistic is shown in Table 4 and the correlation is shown in Figure 2.

### The Path Analysis Model

Maximum-likelihood estimations were used to estimate the path analysis model. Goodness of fit statistics indicated that the model was an adequate representation of the data. (Table 5).

The path analysis model had acceptable model fit (χ^2^/df = 1.894, RMSEA = 0.069, [0.026, 0.10] 90%CI NFI = 0.981, CFI = 0.992). The path diagram of the model was illustrated in Figure 3. In addition, the standardized beta coefficients and *p* value for estimates from the model were presented in Table 6. With the mediator psychache, higher authority vice, higher individualism and higher disposable income of residents significantly predicted less suicidal behavior. Purity vice, collectivism and proportion of the primary industry had positive significant mediating effect on suicidal behavior via the mediator psychache. However, the coefficients of harm vice, fairness vice, ingroup vice, public transport and car for every 10,000 people, urban population density, GDP, urban registered unemployment rate and crude divorce rate did not have the mediating effect.

Furthermore, in order to test whether the mediating effect of the model has cross group stability, in this study, we applied the model to three major economic development belts in China. According to the differences in natural conditions, economic resources, economic development level, transportation conditions, and economic benefits of various regions, China is divided into three major economic zones: the eastern coastal zone, the central zone, and the western zone [34]. If the results of multi-group analysis are not significant, it means economic zones have no effect on the complete mediating effect, and if there are significant differences, it means different economic zones have a moderator effect. The goodness of fit statistics of the multi-group complete mediating effect model were χ^2^/df = 2.509, RMSEA = 0.156, NFI = 0.823, CFI = 0.875. This meant that different economic zones had no influence on the model designed in our study.

## 4. Discussion 

In this study, the path analysis model was designed, and psychache was introduced as the mediation mediate variable. We found that authority vice, purity vice, individualism, collectivism, proportion of the primary industry, and disposable income of residents had mediating effects on suicidal behavior via the mediator psychache. Additionally, the path analysis model showed that harm vice, fairness vice, ingroup vice, public transport and car for every 10,000 people, urban population density, GDP, urban registered unemployment rate, and crude divorce rate had no mediating effect on suicidal behavior. To this end, our results in this study showed that psychache mediated the relationship between social factors (authority vice, purity vice, individualism, collectivism, proportion of the primary industry, and disposable income of residents) and suicidal behaviors.

### 4.1. Cultural/Value Factors

Both collectivism and individualism had indirect effects on suicidal behavior through the mediator psychache, but their effects were completely opposite one another. The results showed that the decrease in collectivism or the increase in individualism in social values could lead to a decrease in psychache, thus leading to the decrease of suicidal behavior. The reason for this result may be that too much emphasis placed on collectivism, especially in China, often means obligation, dedication, obedience, or even sacrifice of personal interests [35]. This collectivism is contrary to the individualistic thought brought about by China’s rapid economic development and urbanization in recent decades [35]. Such conflicts and contradictions often bring negative emotions such as psychache to individuals. Thus, overemphasizing collectivism could lead to more psychache and ultimately to the increase of suicidal behavior, whereas more individualism could reduce psychache and thus reduce suicidal behavior.

Social morality. Authority vice, as a kind of behavior that breaks the hierarchical structure of the group, directly affected suicidal behavior [32]. Its standardized coefficient indicated an indirect negative correlation between authority vice and suicidal behavior. However, the psychology of authority has been studied in terms of fascism and blind obedience [36,37], and the individual who is forced to obey authority often experiences psychological pain [37]. In China, especially in rural areas, family ethics make family members have strong emotional expectations. Once the expectation fails, it can cause a strong emotional reaction to resist [5]. Suicide in China can be regarded as a means of rebelling against authority [38]. Too much emphasis on obedience to parents, the patriarchal clan system, village communities, and other authorities [39] may bring about painful psychological experiences for individuals, who may even commit suicide. Therefore, society pays attention to resisting authority, which encourages individuals to refuse to submit to authority, thus relieving psychache and reducing suicidal behavior. However, resistance to authority is the embodiment of individualism. Individualism resists authority and all attempts to control individuals, especially those imposed by the state or society. Individualism is a protective factor for suicidal behavior (this result has been proved in the previous part). When the population expresses more words resisting authority, it often means the awakening of individual consciousness.

Social morality. Purity vice, as a kind of behavior that suppresses selfishness, is often associated with humanity’s carnal nature (e.g., lust, hunger, material greed) by cultivating a more spiritual mindset [40]. Its standardized coefficient suggested that higher purity vice would lead to more psychache and suicidal behavior. The psychology of purity and disgust has been related to stigma [41]. Stigma has a significant impact on self-esteem and a negative effect on mental health [42]. The frequent mention of purity vice words by an individual may be accompanied by an increase in stigma and negative mental health and a decrease in self-esteem, resulting in psychache and suicidal behavior.

### 4.2. Economic Factors

The indirect effect of disposable income of residents was significant. It revealed that residents’ disposable income could alleviate the occurrence of suicide through the negative effect of psychache. A possible reason is that the lower the income, the lower the self-efficacy, and the greater the probability of negative emotions [43], with negative emotions such as psychache leading to suicidal behavior. Many studies have proved that low-income groups are more likely to commit suicide than high-income groups [44,45,46]. Based on the research above and the analysis of this paper, the mechanism behind the high suicide risk of low-income groups can be explained as follows: The negative emotion psychache caused by high income was reduced, which to some extent hindered the occurrence of suicidal behavior. The influence path of low income was just the opposite. The psychache of low-income groups will often stimulate the emergence of suicidal ideation. This suggests that measures should be taken to alleviate the psychache of low-income populations so as to reduce the possibility of suicide resulting from low income.

### 4.3. Other Factors

The coefficient of proportion of the primary industry to the suicidal behavior through psychache was 0.037. A higher proportion of primary industries indicated a higher suicide rate. This result is supported by existing research. Suicide rate was positively associated with primary industry percentage with a significant tendency [47]. It should be noted that a higher proportion of the primary industry indicated that more people engaged in agricultural activities and lived in the countryside. Rural areas, the main concentration area of the primary industry, were indeed areas with high suicide rates in China from 1995–1999. The suicide rate in rural areas is three times higher than that in urban areas [48]. The decline of the primary industry and the development of the secondary and tertiary industries cause a large agricultural population to flow into the city, and they transform into a non-agricultural population. This process is often accompanied by the improvement of individual income levels [49], the awakening of individual consciousness [50], and less exposure to suicide tools such as pesticides (60% of suicide in China was caused by pesticides) [48]. These factors are caused by the decrease in the proportion of the primary industry and its interaction with others to reduce psychache, thus reducing suicidal behavior.

## 5. Implications

The ultimate goal of the study was to develop a reliable and valid measure that can be used in suicide prevention services. The present study reached the goal of developing a reliable measure. Our study’s results can play a role in implementing the formulation of relevant policies and social development.

### 5.1. Encourage Individualism’s Development in Collectivist Environments

The study’s results indicate that collectivism is a risk factor of suicide and that individualism is a protective factor. China is a typical collectivist country that has shown large effects, being both less individualistic and more collectivistic [51]. Especially to the east of the Hu Huanyong line and to the south of the Qinling Huaihe line, provinces such as Jiangsu, Anhui, Shanghai, Sichuan, and Yunnan are the most collectivistic regions in China [52]. Relevant departments in these areas should seize the opportunity of the growing awakening of individual consciousness and encourage individuals to embrace their independence and initiative in the field of production and life [53]. Specifically, in the areas where collectivism prevails, the government should vigorously improve people’s mental health quality, popularize mental health knowledge, establish community psychological counseling, and improve the mental health care system. This can not only promote collectivism’s development [53] but can also reduce suicidal behavior.

### 5.2. Pay Attention to Primary Industry Practitioners’ Mental Health Problems and Advocate the Reform of the Industrial

The proportion of the primary industry is a risk factor of suicide. This conclusion inspires us to formulate corresponding policies from two perspectives. On the one hand, the government should pay more attention to the mental health of primary industry practitioners. According to our analysis, relevant departments are supposed to pay more attention to the psychological counseling of populations engaged in the primary industry. More specifically, the government can issue a series of policies to encourage the counselors to pay attention to the mental health of the primary industry practitioners, especially the rural farmers, and tilt the psychological resources to the rural areas. In addition, the government can open more consultation hotlines for suicide prevention or involve more knowledge about suicide prevention in the daily cultural propaganda of rural communities.

On the other hand, the government needs to encourage industrial reform so as to reduce the number of practitioners in the primary industry. The local government should vigorously accelerate the upgrading of the primary industry’s industrial structure through scientific and technological innovation and industrial agglomeration [54].The adjustment of the industrial structure of the primary industry is conducive to the adjustment of the employment structure of the primary industry [55,56]. After structural reform, the primary industry can not only reduce the population engaged in the primary industry properly but can also improve production efficiency, reduce suicide, and promote social development.

### 5.3. Pay Attention to Low-Income People’s Mental Health Problems and Improve the Disposable Income of Residents

This study showed that disposable income is a protective factor of suicide. Higher disposable income can alleviate psychache and reduce suicidal behavior. Therefore, relevant departments should pay more attention to the mental health of low-income people and incline more mental resources to avoid the suicide of low-income individuals. For example, we suggest that the government popularize mental health knowledge to low-income groups free of charge and open a free psychological counseling hotline. However, to fundamentally solve suicide caused by low income, it is necessary for local governments to create employment opportunities, promote economic growth [57], reform the tax system [58], improve the financial market and promote the diversification of residents’ property [59], and use other methods to improve residents’ disposable income so as to avoid the negative impact of low disposable income.

### 5.4. Advocate the Social Morality of Pure and Non-Superstitious Authority

Higher purity vice will promote suicide, whereas less authority vice will reduce the occurrence of suicide. The government can formulate policies from the perspective of education, civilizing people’s customers, and strengthening the legal system to promote the establishment of a pure and non-superior society [60].

## 6. Limitations 

Although there are important discoveries revealed by this research, there are also limitations. First, the data selected are biased. Users of Sina Weibo tend to be younger, so this study ignores the suicidal behavior of elderly and non-microblog users. Thus, there may be some bias in the results. Second, it is also worth noting that some independent variables that are considered to be related to suicidal behaviors were not significant in the analysis. Perhaps they are not significant because our research was mainly based on Shneidman’s theory [12], which regards psychache (negative hurt, mental pain, soreness) as the cause of all suicidal behaviors. Thus, psychache was the only mediator in our model. It simplified the psychological mechanism. Therefore, it is not comprehensive to attribute all social factors influencing suicidal behavior only through psychache. In reality, the psychological mechanism of suicidal behavior may be more complex, which requires further study.

## 7. Conclusions 

We used the path analysis model to analyze the mediating mechanism of social factors on suicidal behavior. We applied our model to three major economic development belts in China to test whether the mediating effect had differences. Our study found that harm vice, fairness vice, ingroup vice, public transport and car for every 10,000 people, urban population density, GDP, urban registered unemployment rate, and crude divorce rate had no significant mediating effect on suicidal behavior. Authority vice, purity vice, individualism, collectivism, proportion of the primary industry, and disposable income of residents had significant mediating effects on suicidal behavior via the mediator psychache, and the mediating effect of our model had no differences in different areas of China. These results bring new inspiration to our prevention of suicidal behavior at the social level. Future research needs to test whether more diverse populations meet the results of this study.

## Figures and Tables

**Figure 1 ijerph-18-02604-f001:**
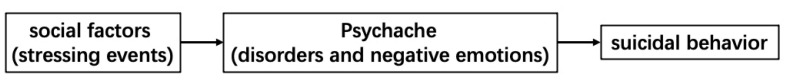
Path analysis model designed.

**Figure 2 ijerph-18-02604-f002:**
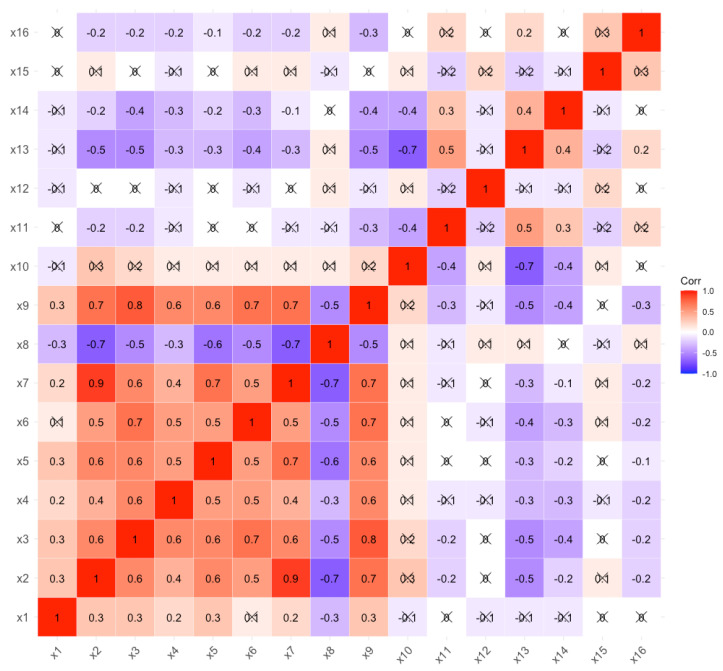
Thermodynamic diagram of correlation coefficient. Note: ×1-Suicidal behavior, ×2-Psychache, ×3-Harm vice, ×4-Fairness vice, ×5-Ingroup vice, ×6-Authority vice, ×7-Purity vice, ×8-Individualism, ×9-Collectivism, ×10-Proportion of the primary industry, ×11-Public transport and car for every 10,000 people, ×12-Urban population density, ×13-Disposable income of residents, ×14-Gross domestic product (GDP), ×15-Urban registered unemployment rate, ×16-Crude divorce rate; “×”represents the correlation between variables is not significant (p<0.05).

**Figure 3 ijerph-18-02604-f003:**
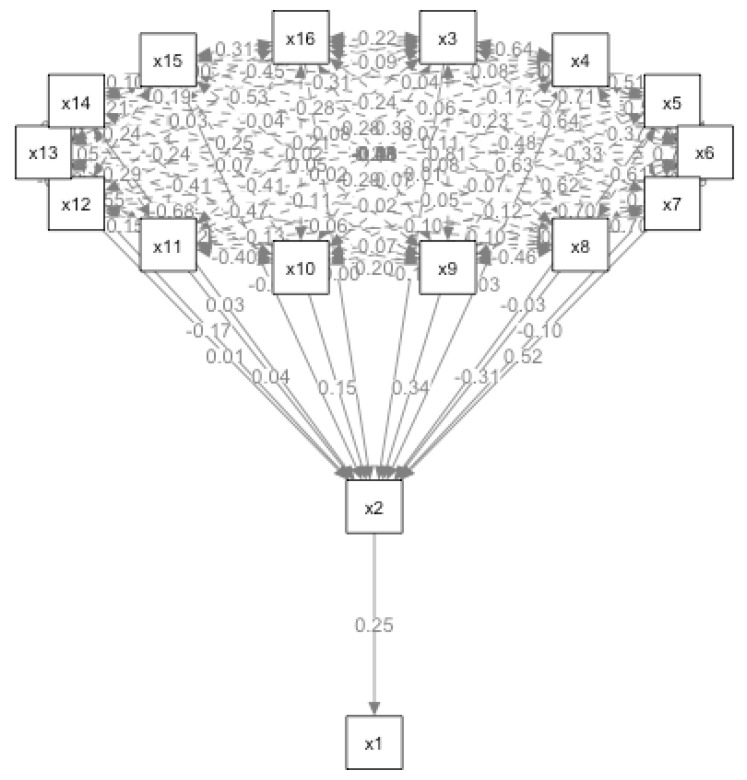
Path analysis model. Note: ×1-Suicidal behavior, ×2-Psychache, ×3-Harm vice, ×4-Fairness vice, ×5-Ingroup vice, ×6-Authority vice, ×7-Purity vice, ×8-Individualism, ×9-Collectivism, ×10-Proportion of the primary industry, ×11-Public transport and car for every 10,000 people, ×12-Urban population density, ×13-Disposable income of residents, ×14-Gross domestic product (GDP), ×15-Urban registered unemployment rate, ×16-Crude divorce rate.

**Table 1 ijerph-18-02604-t001:** Chinese suicide dictionary.

Category	Definition	Number of Words	Representative Words
Suicide ideation	Words reflecting suicidal thoughts	586	want to die escape
Suicidal behavior	Words reflecting self-harm behaviors	88	seppuku hypnotics
Psychache	Words reflecting psychological distress	403	want to cry loneliness
Mental illness	Words reflecting poor mental health status	48	depression hallucination
Hopeless	Words reflecting a feeling of despair	188	dead end despair
Somatic complaints	Words reflecting somatic symptoms	183	headache shortness of breath
Self-regulation	Words reflecting an attempt to push oneself hardly	36	repression force oneself to smile
Personality	Words reflecting negative personality inferiority	72	complex hate oneself
Stress	Words reflecting pressure in daily life	83	failure pressure
Trauma/hurt	Words reflecting traumatic or unpleasant experiences	182	get dumped infidelity
Talk about others	Words reflecting one’s relatives and friends	47	partner son
Shame/guilt	Words reflecting a feeling of shame and guilt	72	lose status making an apology
Anger/hostility	Words reflecting a feeling of angry and hostile against others	180	damn it curse

**Table 2 ijerph-18-02604-t002:** Cultural value dictionary.

Category	Definition	Number of Words	Representative Words
Individualism	Words reflecting individualism	53	Independence autonomy
Collectivism	Words reflecting collectivism	64	Team cooperation

**Table 3 ijerph-18-02604-t003:** Moral foundation dictionary.

Category	Definition	Number of Words	Representative Words
Harm Virtue	Words reflecting care	40	Care friendship
Harm Vice	Words reflecting hurt	81	Torture kill
Fairness Virtue	Words reflecting fairness	38	Decent equal
Fairness Vice	Words reflecting injustice	34	Fraud injustice
Ingroup Virtue	Words reflecting unity	72	Motherland unity
Ingroup Vice	Words reflecting rebellion	53	Treason heresy
Authority Virtue	Words reflecting authority	83	compliance
Authority Vice	Words reflecting revolt	48	demonstration
Purity Virtue	Words reflecting purity	60	loyalty chastity
Purity Vice	Words reflecting filthiness	101	Filth sin
Morality General	Words reflecting other general morality not mentioned	73	Praise value

**Table 4 ijerph-18-02604-t004:** Descriptive statistics for variables (2013–2018).

Variable			Data Sources	Mean $\left(10^{-4}\right)$	SD $\left(10^{-5}\right)$
	Dependent variable	Suicidal behavior	Chinese suicide dictionary/ Sina Weibo	2.00	1.78
	Mediating variables	Psychache	Chinese suicide dictionary/ Sina Weibo	44.29	101.01
Cultural/Value factors	Individualism and collectivism	Individualism	Cultural Value Dictionary/ Sina Weibo	219.30	436.29
		Collectivism	Cultural Value Dictionary/ Sina Weibo	76.74	60.46
	Social morality	Harm vice	Moral Foundation Dictionary/ Sina Weibo	3.07	4.09
		Fairness vice	Moral Foundation Dictionary/ Sina Weibo	0.61	0.83
		Ingroup vice	Moral Foundation Dictionary/ Sina Weibo	1.18	1.96
		Authority vice	Moral Foundation Dictionary/ Sina Weibo	1.02	1.41
		Purity vice	Moral Foundation Dictionary/ Sina Weibo	4.23	19.46
Economic factors	Disposable income of residents	Disposable income of residents (yuan)	National Bureau of Statistics	230,291,985	1,000,441,740
	Gross domestic product	GDP (million yuan)	National Bureau of Statistics	246,386,565	1,974,809,840
Other factors	Urbanization	Proportion of the primary industry	National Bureau of Statistics	957.67	4974.43
		Public transport and car for every 10,000 people (unit)	National Bureau of Statistics	129,440.32	317,058.75
		Urban population density (people per square kilometer)	National Bureau of Statistics	28,239,193.5	112,142,778
	Unemployment	Urban registered unemployment rate (%)	National Bureau of Statistics	32,327.96	64,073.42
	Marriage	Crude divorce rate (‰)	National Bureau of Statistics	2.934	0.96

**Table 5 ijerph-18-02604-t005:** Goodness of fit statistics of path analysis model.

Path Analysis Model Fit				
χ^2^/df (<3)	RMSEA (<0.08)	RMSEA 90%CI	NFI (>0.9)	CFI (>0.9)
1.894	0.069	[0.026, 0.10]	0.936	0.967

**Table 6 ijerph-18-02604-t006:** Results of the path analysis model.

Path	Standardized Estimate	*p* (>|z|)
***Psychache***		
psychache → Suicidal behavior	0.252	0.000 ***
**Cultural/Value factors**		
***Individualism and collectivism***		
Individualism → psychache → Suicidal behavior	−0.077	0.000 ***
Collectivism → psychache → Suicidal behavior	0.086	0.000 ***
***Social morality***		
Harm vice → psychache → Suicidal behavior	−0.028	0.055
Fairness vice → psychache → Suicidal behavior	−0.007	0.477
Ingroup vice → psychache → Suicidal behavior	−0.007	0.520
Authority vice → psychache → Suicidal behavior	−0.026	0.014 **
Purity vice → psychache → Suicidal behavior	0.130	0.000 ***
**Economic factors**		
Disposable income of residents		
Disposable income of residents → psychache → Suicidal behavior	−0.044	0.000 ***
Gross domestic product (GDP)		
GDP → psychache → Suicidal	0.008	0.362
**Other factors**		
***Urbanization***		
Proportion of the primary industry → psychache → Suicidal behavior	0.037	0.000 ***
Public transport and car for every 10,000 people → psychache → Suicidal behavior	0.011	0.214
Urban population density → psychache → Suicidal behavior	0.004	0.637
***Unemployment***		
Urban registered unemployment rate → psychache → Suicidal	−0.004	0.660
***Marriage***		
Crude divorce rate → psychache → Suicidal behavior	−0.0002	0.975

Note: ** *p* < 0.05, *** *p* < 0.01.

## Data Availability

Restrictions apply to the availability of these data. Data was obtained from Weibo and are available at (https://weibo.com) with the permission of Weibo.

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
