# Peer review of "Influential Path of Social Risk Factors toward Suicidal Behavior—Evidence from Chinese Sina Weibo Users 2013–2018"

_ijerph, 2021, doi:10.3390/ijerph18052604_

Round 1

Reviewer 1 Report

Dear authors,

Thanks for submitting this paper for consideration in Int. J. Envi-ron. Res. Public Health. After a careful review of your paper, I have some comments to provide, in order to help strengthening the form and contents of this study.

First of all, the topic covered by the study is worth of investigation, and deserves an empirical approach, such as done by the authors. Further, more powerful and complex models are needs, in order to develop better hypotheses on the factors potentially influencing suicidal attempts and their related consequences. In this regard, authors have done a good effort, providing a potentially suitable framework for it.

There are some key issues that, however, need more work, and some others require major improvements, as I will list below:

Authors should redesign several parts of the abstract. In its current form, it seems overcharged and offers not too much information on key issues, such as the data analyzed. These data, rather than redundant information that is later repeated at the text, could be more useful for readers.

Also, authors report goodness of fit indicators of the multi group model at the abstract. I really hesitate on the fact that this is strictly necessary, and if whether just qualitatively commenting on these fit indexes (that have some limitations needed to be discussed afterwards) could be better to simplify this section.

Further, I found inadequate the statement about that the results, that “bring new inspiration” to prevent suicidal behavior. Although this is true, this section of the abstract is intended to formulate implications based on the data and the relationships observed in the study.

As for the literature review, and although the authors done a good addressing of all main components of their study, used as potential predictors, the mechanisms linking them (integrating these factors into a scientifically plausible set of hypotheses theoretically supported) seem not to be properly formulated by the authors.

The aforementioned also affects the support given to the study aim; if the way by which these factors are possibly connected is not explained, the aim seems week. Of course, this requires a careful revision from the authors.

As previously stated, the data used should be well described. In its current form, it is very difficult to differentiate key elements such as cases, data sources, codes, demographic factors, study variables and how were the procedures done during the study. If you consider that reviewers struggle doing so, just think about non-specialized readers of the paper.

Figure 1, although interesting, lacks from any value, given that the structure presented is not properly described in the text. It is too similar to Figure 2 (that includes some estimates that, anyway, should be better presented/edited, since they are quite illegible), and does not add several highlights to the paper.

The goodness of fit of the path model needs more discussion, addressing the limitations of having (e.g.) a slightly high RMSEA, bringing theoretical support to the model fit, and not only stating it was acceptable.

Discussion is interesting, and has been segmented in a good form, so that it is possible to have an idea on the different categories of the variables in regard to suicidal behavior.

In regard to study limitations, there are many of them that have been ignored or underdiscussed by the authors. This section, especially, requires further amendments from the authors.

The conclusions are interesting, but the practical implications could be strengthened and improved, working on the synthesis of the results and the way they could be implied in potential interventions or policies.

Author Response

Thanks for your comments. Please see the attachment for more reply.

Reviewer 2 Report

Thank you very much for asking me to review the present manuscript.

This is, in summary, an interesting study aimed to determine suicidal risk factors, and the relation and underlying mechanism between social variables and Sina Weibo users’ suicide behavior. The authors reported that the path analysis model was an adequate representation of the data. In addition, by mediator psychache, higher authority vice, individualism and disposable income of residents significantly predicted less suicide behavior. Purity vice, collectivism and proportion of the primary industry had positive significant effect on suicide behavior via the mediator psychache as well. Moreover, different economic zones had no influence on the designed model.

The authors may find my minor comments below.

First, when throughout the Introduction section, the authors correctly focused on the disability and psychosocial impairment related to suicideal behavior, the link between depression and negative outcomes (e.g., suicidal behavior) might be further discussed. Importantly, an association between depression, and suicidality has been repeatedly reported. Overall, those who attempt or complete suicide are characterized by mood disorders, stressful life events, interpersonal problems, and feelings of hopelessness. Thus, in order to briefly discuss this topic (although i understand that the link between depression and suicidal behavior is not the main topic of the present manuscript), i suggest to cite, within the main text, the paper published in 2012 on Curr Pharm Des (PMID: 22716157).

Furthermore, the Introduction section is too long and needs to be reduced in length for the general readership.

Moreover, the main rationale based on which the authors used individualism and collectivism dictionary to calculate the word frequencies of individualism and collectivism published on Sina Weibo need to be specified.

Also, the authors could immediately present and discuss, both in the first lines of the Discussion and Conclusion sections, the most relevant study findings of this paper instead of focusing on the most relevant aims/objectives that should have been stressed elsewhere within the main text

Finally, what is the take-home message of the present manuscript? While the authors reported that authority vice, purity vice, individualism, collectivism, proportion of the primary industry, and disposable income of residents had significant mediating effects on suicide behavior, they failed, in my opinion, to provide their point of view abot the main topic. Specifically, which type of new inspiration for prevention of suicide behavior the authors suggested? Which type of mediators need to be further discussed and explored in a detailed manner? Here, more details/information are required according to the authors’ expertise.

Author Response

(The authors gave the same response as above.)

Round 2

Reviewer 1 Report

Dear authors,

Thanks so much for the responses, rationales and amendments provided in the revised version of your manuscript. After a careful (second) read, my impression is that the scientific soundness and value for readers are greater thanks to the good efforts you put in the revisions and clarifications.

Considering the above mentioned, I believe this revised version of the paper can be accepted for publication.

Best wishes and thanks for the improvements submitted.

Reviewer 2 Report

In the revised manuscript, the authors addressed sufficiently most of the major comments raised by Reviewers. I have no further comments.